# Selective emergence of photoluminescence at telecommunication wavelengths from cyclic perfluoroalkylated carbon nanotubes

Yutaka Maeda [1✉], Yasuhiro Suzuki[1], Yui Konno[1], Pei Zhao [2✉], Nobuhiro Kikuchi[1], Michio Yamada [1], Masaya Mitsuishi [3], Anh T. N. Dao [4,5], Hitoshi Kasai[4] & Masahiro Ehara [2✉]

Chemical functionalisation of semiconducting single-walled carbon nanotubes (SWNTs) can tune their local band gaps to induce near-infrared (NIR) photoluminescence (PL). However, tuning the PL to telecommunication wavelengths (>1300 nm) remains challenging. The selective emergence of NIR PL at the longest emission wavelength of 1320 nm was successfully achieved in (6,5) SWNTs via cyclic perfluoroalkylation. Chiral separation of the functionalised SWNTs showed that this functionalisation was also effective in SWNTs with five different chiral angles. The local band gap modulation mechanism was also studied using density functional theory calculations, which suggested the effects of the addenda and addition positions on the emergence of the longest-wavelength PL. These findings increase our understanding of the functionalised SWNT structure and methods for controlling the local band gap, which will contribute to the development and application of NIR light-emitting materials with widely extended emission and excitation wavelengths.

[1] Department of Chemistry, Tokyo Gakugei University, Tokyo 184-8501, Japan. [2] Research Center for Computational Science, Institute for Molecular Science, Okazaki 444-8585, Japan. [3] Graduate School of Engineering, Tohoku University, Sendai 980-8579, Japan. [4] Institute of Multidisciplinary Research for Advanced Materials (IMRAM), Tohoku University, Sendai 980-8577, Japan. [5] Graduate School of Engineering, Nagasaki University, Nagasaki 852-8521, Japan. ✉email: ymaeda@u-gakugei.ac.jp; pei@ims.ac.jp; ehara@ims.ac.jp

Single-walled carbon nanotubes (SWNTs), which have a tubular graphene sheet structure, are nanocarbon materials with high mechanical strength owing to the carbon-carbon bonds and an interesting electronic structure owing to their one-dimensionally expanded cylindrical π-electron system[1–4]. The chemical and physical properties of SWNTs depend on structural parameters such as the diameter and angle of the six-membered ring lattice. The chiral index $(n,m)$, which represents the graphene vector of the development view of SWNTs, has been used as an excellent nomenclature to uniquely describe the SWNT structure. SWNTs with chiral indices of $(n,m)$ are hereinafter referred to as $(n,m)$ SWNTs. SWNTs with a chiral index of $mod(n\text{-}m,3) = 0$ are metallic; otherwise, they are semiconducting. Since the near-infrared (NIR) photoluminescence (PL) of semiconducting SWNTs was reported in 2002[5,6], the NIR PL properties have attracted attention not only as probes for the assignment of their chiral index but also as an NIR light source for bioimaging[7], sensors[8], telecommunications[9,10], and other applications. Because of the rigid structure of SWNTs, the Stokes shift of SWNTs is small; therefore, it requires excitation with the second-lowest excitation energy ($E_{22}$) for the PL ($E_{11}$ PL) observation. Additionally, increasing the low PL quantum yield of SWNTs is significant for their application[11]. Chemical functionalisation can reportedly tune the band gap locally, resulting in the emergence of new PL peaks at a longer wavelength than the $E_{11}$ PL wavelength[12–14]. The PL efficiencies of both intrinsic PL and PL that emerges by functionalisation are sensitive to the functionalisation degree[15–18]. For example, arylated (6,5) SWNTs with an appropriate functionalisation degree achieve a 28-fold higher PL efficiency than pristine SWNTs[17]. The sufficient energy difference between the PL wavelength created by functionalisation and the lowest excitation wavelength ($E_{11}$) allows $E_{11}$ excitation, which increases both the excitation and emission efficiency at the appropriate functionalisation degree.

Light sources emitting at wavelengths above 1300 nm are suitable for optical communications. Three strategies have been proposed to control the PL wavelength of SWNTs by functionalisation, namely, control of the binding configurations[19,20], electronic effects of the addenda[17,21], and local strain at the binding sites[22]. The importance of the binding configurations on the local band gap has been studied using theoretical calculations, and their influence is significantly larger than that of the other parameters. For example, Htoon et al. reported the low-temperature PL spectroscopy of 3,5-dichlorobenzene-functionalised (6,5) SWNTs, which exhibited multiple sharp PL peaks from 1000 to 1350 nm[23]. These PL peaks were assigned to the 1,2- and 1,4-isomers based on theoretical calculations of model compounds with distinct aryl-H binding configurations using time-dependent density functional theory (TD-DFT). Therefore, by comparing the experimental and theoretical calculation results, it has been proposed that the PL wavelength depends on the binding configurations of the additive. Substituent electronic effects on functionalised SWNTs were reported by Wang et al. They demonstrated that the PL wavelength was controlled in the range of 1110–1137 nm by the *para*-substituent effect of phenylated (6,5) SWNTs[17], and that in the range of 1096–1155 nm was controlled by replacing the hydrogen atoms of the hexylated SWNTs with fluorine atoms[20]. Maeda et al. reported that the functionalisation of (6,5) SWNTs using 1,n-dibromoalkane ($BrC_nH_{2n}Br$, $n = 3$–5), a reagent with two reactive sites, resulted in the selective emergence of a new PL peak at 1215–1231 nm[22]. In contrast, functionalisation using 1-bromoalkane ($C_nH_{2n+1}Br$, $n = 1$–4) instead of 1,n-dibromoalkane resulted in two new PL peaks at 1100–1230 nm[24]. 1,3-dibromopropane and 1,4-dibromobutane have been used to synthesise the cyclisation products of fullerene $C_{60}$[25].

Based on these facts, it is expected that new PL peaks will selectively emerge at longer wavelengths than previously reported by combining the electronic effects of the substituent and controlling the addition position. No methods exist to selectively emerge new PL peaks >1300 nm from (6,5) SWNTs on the bulk scale[26–28]. In this study, SWNT functionalisation using 1,4-diiodooctafluorobutane, which incorporates two reactive sites and fluorine atoms, was investigated to further shift the PL to telecommunications wavelengths, and the unique reactivity and PL characteristics were clarified.

## Results and discussion

### Effect of fluorine atoms on the functionalisation degree of functionalised (6,5) SWNTs.

SWNTs were functionalised using 1,4-diiodooctafluorobutane ($I(CF_2)_4I$) and sodium naphthalenide. High functionalisation efficiency was expected owing to the six-membered ring structure of the resulting cycloadducts[29]. To clarify the effects of the two reactive sites and number of fluorine atoms, 1-iodobutane derivatives (RI, $R = (CH_2)_3CH_3$, $(CH_2)_3CF_3$, $(CH_2)_2CF_2CF_3$, $CH_2(CF_2)_2CF_3$, $(CF_2)_3CF_3$) and 1,4-diiodo-2,2,3,3-tetrafluorobutane ($ICH_2(CF_2)_2CH_2I$) were used instead of $I(CF_2)_4I$; hereafter, the functionalised SWNTs are designated as SWNT-R, SWNT>$CH_2(CF_2)_2CH_2$, and SWNT>$(CF_2)_4$, respectively. Figure 1 and Supplementary Figs. 1, 2 show the absorption spectra and Raman spectra of the functionalised SWNTs at an excitation wavelength of 561 nm. An increase in the D band (~1300 cm$^{-1}$) toward the G band (~1590 cm$^{-1}$) and a decrease in the characteristic absorption peaks, which are characteristic features of sidewall functionalisation, were observed[30] (Supplementary Data 1). The addition of fluoroalkyl groups was also supported by the detection of fluorine atoms using X-ray photoelectron spectroscopy (Supplementary Fig. 3). Interestingly, the D/G values after functionalisation increased in the order: [SWNT>$CH_2(CF_2)_2CH_2$] < [SWNT-$CH_2(CF_2)_2CF_3$] < [SWNT-$(CF_2)_3CF_3$] < [SWNT>$(CF_2)_4$] < [SWNT-$(CH_2)_2CF_2CF_3$] < [SWNT-$(CH_2)_3CF_3$] < [SWNT-$(CH_2)_3CH_3$] < [SWNT>$(CH_2)_4$][22,29]. With the exception of SWNT-$CH_2(CF_2)_2CF_3$ and SWNT>$CH_2(CF_2)_2CH_2$, the D/G values after functionalisation tended to decrease with an increasing number of fluorine atoms. These results indicate that the substitution of hydrogen atoms close to the reactive site with fluorine atoms suppressed the addition reaction of the SWNTs. Additionally, $Br(CH_2)_4Br$ and $I(CF_2)_4I$, with two reactive sites, exhibited higher reactivity than $CH_3(CH_2)_3I$ and $CF_3(CF_2)_3I$. Regarding the effect of halogen atoms at the reactive sites, the D/G values of SWNT-$(CH_2)_3CH_3$ synthesised using 1-iodobutane and 1-bromobutane were 0.21 and 0.19, respectively, indicating that iodoalkane is more reactive than the corresponding bromoalkane.

### Effect of fluorine atoms on the PL characteristics of functionalised (6,5) SWNTs.

Figure 1c depicts the PL spectra of functionalised SWNTs dispersed in a 1 wt% sodium dodecylbenzenesulfonate (SDBS) $D_2O$ solution, in which the excitation wavelengths are suitable for the excitation of (6,5) SWNTs. There was a good correlation between the changes in the $E_{11}$ absorption peak intensity and the $E_{11}$ PL peak intensity (Supplementary Fig. 4). SWNT-$(CH_2)_3CF_3$ (1104 and 1240 nm) and SWNT-$(CH_2)_2CF_2CF_3$ (1118 and 1243 nm) exhibited two new PL peaks (namely, $E_{11}^*$ and $E_{11}^{**}$ PL), which are similar to SWNT-$(CH_2)_3CH_3$ (1100 and 1231 nm)[24] (Supplementary Table 1). In contrast, one prominent PL peak attributed to the functionalised (6,5) SWNTs was observed from SWNT-$CH_2(CF_2)_2CF_3$ (1119 nm), SWNT-$(CF_2)_3CF_3$ (1152 nm), SWNT>$(CH_2)_4$ (1230 nm)[22,29], SWNT>$CH_2(CF_2)_2CH_2$ (1246 nm), and SWNT>$(CF_2)_4$ (1318 nm). Notably, SWNT>$(CF_2)_4$ selectively emerged a new PL peak at 1318 nm. The PL wavelength of the functionalised (6,5) SWNTs was significantly longer than that of the corresponding alkylated SWNTs, SWNT>$(CH_2)_4$[22,29].

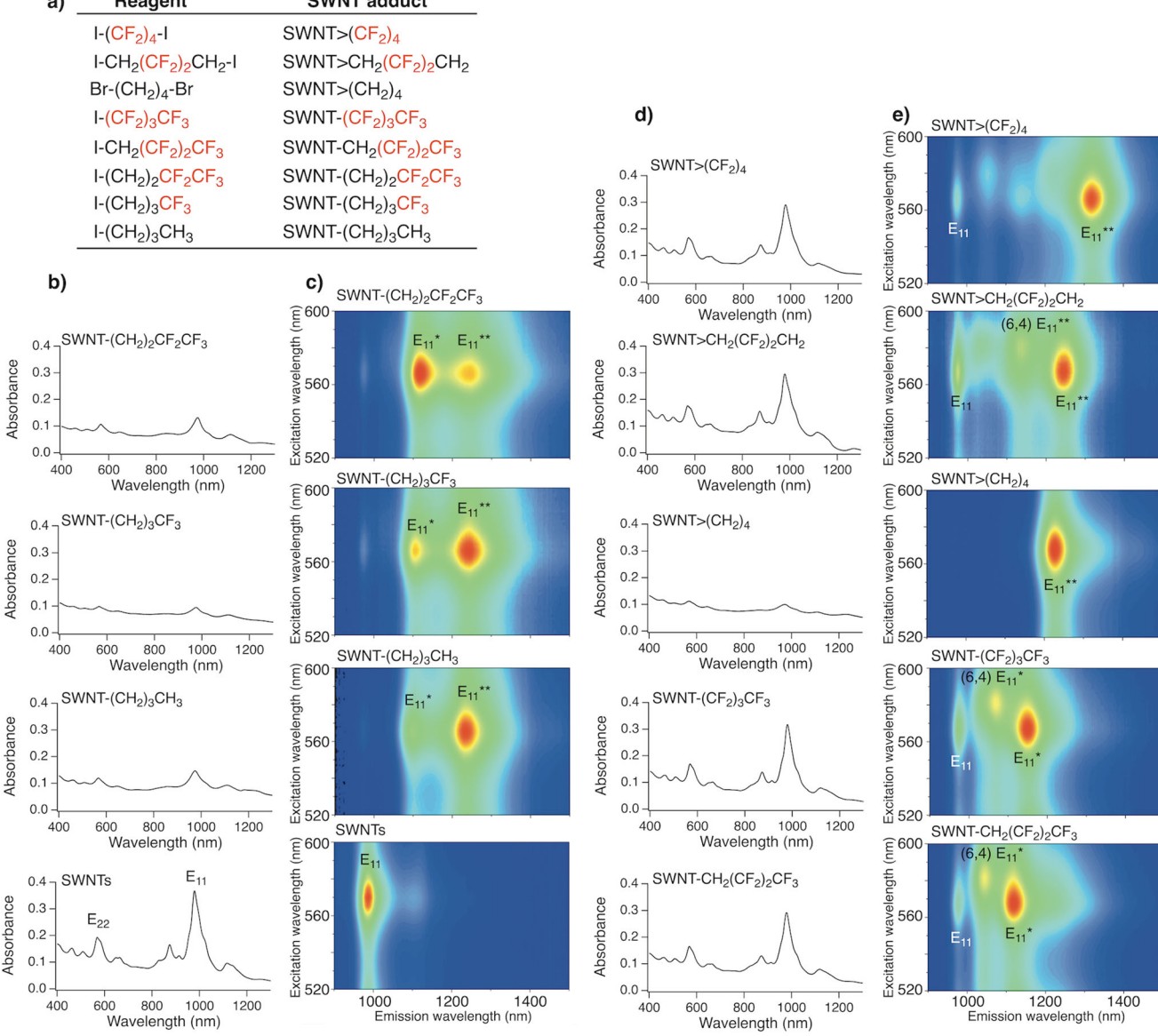

**Fig. 1 Absorption and PL spectra of functionalised SWNT adducts. a** Chemical formulae of the reagents and abbreviations of the corresponding SWNT adducts. **b**, **d** Absorption spectra and **c**, **e** PL mapping of SWNTs and functionalised SWNTs with increasing numbers of fluorine substituents dispersed in $D_2O$ containing 1 wt% SDBS. The peak assignment for (6,5) SWNT ($E_{22}$ and $E_{11}$ absorption, $E_{11}$, $E_{11}^*$, and $E_{11}^{**}$ PL) is shown unless otherwise stated.

The small PL peaks observed at 1043 nm in SWNT-$CH_2(CF_2)_2CF_3$, 1073 nm in SWNT-$(CF_2)_3CF_3$, and 1137 nm in SWNT>$CH_2(CF_2)_2CH_2$ were assigned to the functionalised (6,4) SWNTs, because the wavelength maxima of their excitation spectra matched the $E_{11}$ and $E_{22}$ absorption wavelength of (6,4) SWNTs (Supplementary Figs. 5 and 6)[6]. The number of PL peaks in (6,5) SWNTs emerged by functionalisation strongly depended on the number of fluorine atoms and reactive sites in the reagent. It is noteworthy that, owing to their high selectivity, $CF_3(CF_2)_2CH_2I$, $CF_3(CF_2)_3I$, $Br(CH_2)_4Br$[29], $ICH_2(CF_2)_2CH_2I$, and $I(CF_2)_4I$ produced a single $E_{11}^*$ or $E_{11}^{**}$ PL peak, whereas the other reagents produced two PL peaks. The high selectivity of the PL wavelength suggested that a regioselective reaction occurred. Increasing the number of fluorine atoms shifted both the $E_{11}^*$ and $E_{11}^{**}$ PL wavelengths to longer wavelengths, similar to that of previously reported results[20]. The gradual shift of the $E_{11}^*$ and $E_{11}^{**}$ PL wavelengths, depending on the number of fluorine atoms instead of the difference in the binding sites, results in a PL wavelength change.

**Effect of fluorine atoms on the PL characteristics of functionalised SWNTs with different chiral angles.** The PL contour plots of SWNT>$(CF_2)_4$ exhibited new PL peaks at 1052 and 1219 nm with an excitation wavelength of 869 nm, suggesting that $I(CF_2)_4I$ is effective for significantly decreasing the local band gap of (6,4) SWNTs (Supplementary Fig. 6)[12]. To clarify the effect of controlling the PL properties on SWNTs other than (6,5) SWNTs, gel chromatography was conducted on SWNT>$(CF_2)_4$, SWNT>$CH_2(CF_2)_2CH_2$, and SWNT-$(CF_2)_3CF_3$. The functionalised SWNTs were dispersed in 0.5 wt% sodium dodecylsulfate (SDS) and 0.5 wt% sodium cholate (SC) aqueous solutions and the dispersion was injected into sephacryl gel. The adsorbed SWNTs on the gel were separated using gradient high-performance liquid chromatography (HPLC) using three surfactant mixed systems developed by ref. [31]. The chiral index of the SWNTs, including each fraction that was eluted depending on the sodium deoxycholate (DOC) concentration, was assigned using the absorption spectra (Fig. 2 and Supplementary Figs. 7–9). The solvent of the fraction was then exchanged with the 1 wt% SC

$D_2O$ solution to measure the PL. Hereafter, the separated SWNTs are designated as SWNT-R with the DOC concentration in parentheses. The separated SWNT>$(CF_2)_4$, SWNT>$CH_2(CF_2)_2CH_2$, and SWNT-$(CF_2)_3CF_3$ were assigned to the characteristic $E_{11}$ and $E_{22}$ absorption peaks. For example, separated SWNT>$(CF_2)_4$ exhibited the characteristic absorption peaks assigned to (6,4), (7,3), (9,1), (6,5), (8,3), and (7,5) SWNTs, depending on the DOC

concentration, respectively (Fig. 2b). In particular, the purity of the (6,4), (6,5), and (8,3) SWNTs adducts was high because of the lack of characteristic absorption peaks assigned to the other SWNTs. The trend of the chiral index of the eluted SWNTs adducts at each DOC concentration was in good agreement with those reported in previous studies[20,29,31,32]. The small influence of the addenda on the separation behaviour may be

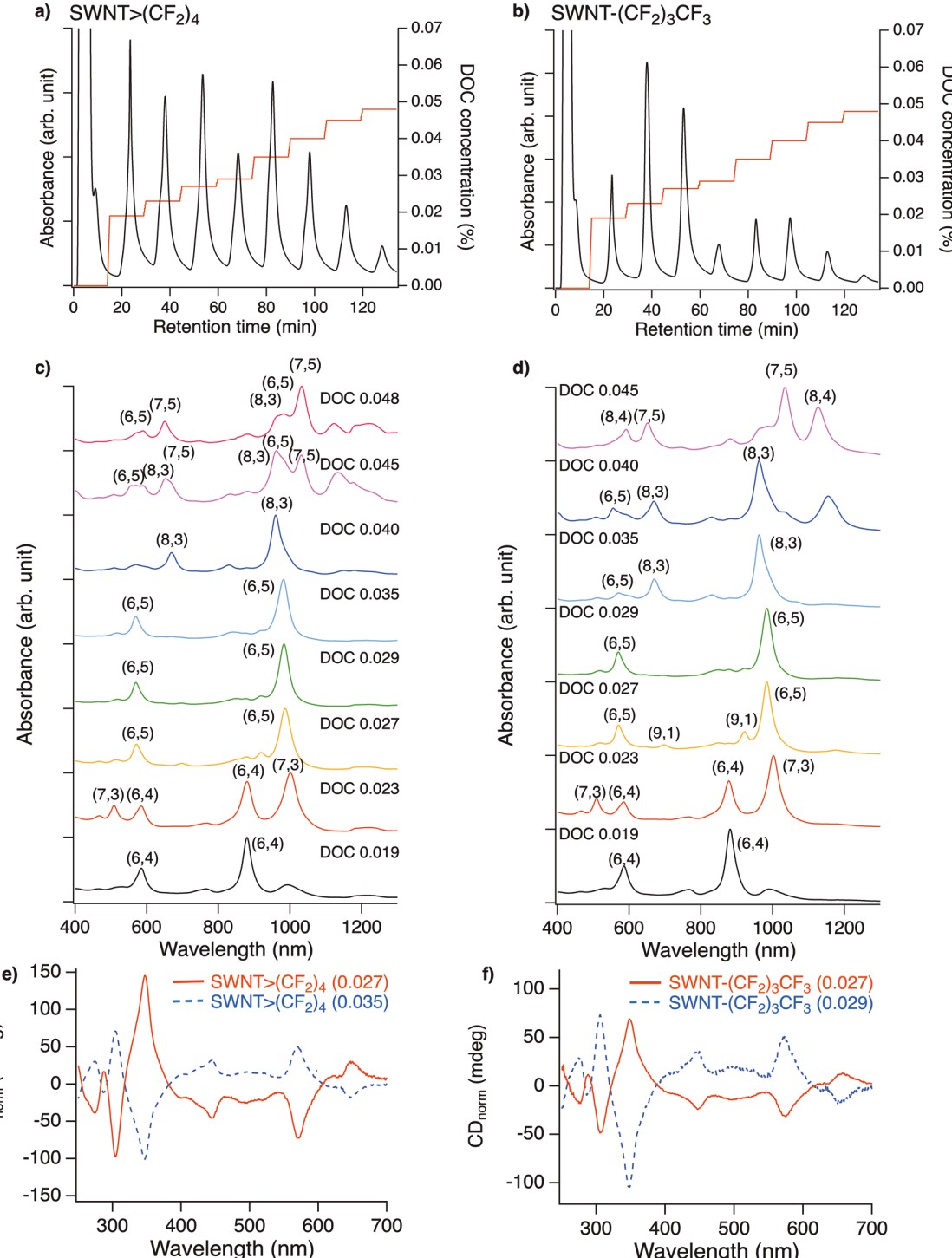

**Fig. 2 Separation and assignment of functionalised SWNTs.** HPLC profile of (**a**) SWNT>$(CF_2)_4$ and (**b**) SWNT-$(CF_2)_3CF_3$ monitored at 280 nm and gradient proportion of DOC concentration (wt%). Condition: column, $\phi10 \times 200$ mm; eluent, $H_2O$ containing 0.5 wt% SC, 1.0 wt% SDS, and X wt% DOC, where X corresponds to the values shown on the gradient; flow rate, 2 mL min$^{-1}$. Absorption spectra of separated **c** SWNT>$(CF_2)_4$ and **d** SWNT-$(CF_2)_3CF_3$ in $D_2O$ containing 1 wt% SC. CD spectra of separated **e** SWNT>$(CF_2)_4$ and **f** SWNT-$(CF_2)_3CF_3$ (right) in $D_2O$ containing 1 wt% SC normalised by the $E_{22}$ absorbance.

due to the low functionalisation degree of SWNT>$(CF_2)_4$, SWNT>$CH_2(CF_2)_2CH_2$, and SWNT-$(CF_2)_3CF_3$. Characteristic absorption peaks assigned to (6,5) SWNTs were observed for SWNT>$(CF_2)_4$ (0.027), SWNT>$(CF_2)_4$ (0.035), SWNT-$(CF_2)_3CF_3$ (0.027), and SWNT-$(CF_2)_3CF_3$ (0.029). The radial breathing mode peaks of the separated fractions also support the assignment of the chiral index of the SWNTs adducts (Supplementary Fig. 10). The circular dichroism (CD) spectra of these fractions exhibited mirror-image CD spectra similar to those of (11,−5) and (6,5) SWNTs, indicating the achievement of optical resolution for the functionalised (6,5) SWNTs (Fig. 2c).

Figure 3 shows the PL contour map for each separated functionalised SWNT. One predominant PL peak emerged after the functionalisation of each separated SWNT>$(CF_2)_4$, SWNT>$CH_2(CF_2)_2CH_2$, SWNT>$(CH_2)_4$[29], and SWNT-$(CF_2)_3CF_3$ (Supplementary Figs. 11–18 and Supplementary Table 2). As an exception, (6,4) SWNT>$(CF_2)_4$ (0.019) and (7,3) SWNT>$(CF_2)_4$ (0.023) exhibited two PL peaks ((6,4) SWNT: 1064 and 1222 nm and (7,3) SWNT: 1373 and 1389 nm). Interestingly, two PL peaks emerged in (6,4) and (7,3) SWNT>$(CF_2)_4$; however, the chirality dependence on the selectivity is currently unclear. Unlike functionalisation with bromobutane, which results in two new PL peaks, it is noteworthy that functionalisation using $CF_3(CF_2)_3I$ exhibited high selectivity to emerge a single new PL peak. The PL peaks of SWNT-$(CF_2)_3CF_3$ were observed at longer wavelengths than those of the corresponding SWNT-$(CH_2)_3CH_3$ regardless of the chiral index, and the PL wavelength of SWNT-$(CF_2)_3CF_3$ was similar to that of the SWNT-$(CF_2)_5CF_3$ reported by Wang et al. (Table 1)[21]. The new PL peak of SWNT>$(CF_2)_4$ was observed at the longest wavelength among those of SWNT>$(CH_2)_4$[29], -$(CF_2)_3CF_3$, and -$(CH_2)_3CH_3$[20], regardless of the chiral index.

The results demonstrate the effectiveness of functionalisation using fluoroalkanes with two reactive sites to control the PL of SWNTs in terms of selectivity and significant wavelength shifting.

Wang et al. reported that there is a linear relationship between the PL energy difference before and after arylation and the diameter multiplier[17]. Figure 4 shows the PL energy differences before and after functionalisation (ΔPL) plotted as a function of the reciprocal diameter ($1/d^2$) of the functionalised SWNTs. The ΔPL tended to increase with an increasing $1/d^2$ for each functionalisation, indicating that functionalisation of SWNTs with a smaller diameter is more influential in the local band gap modulation. The difference in ΔE between SWNT-$(CH_2)_3CH_3$ and SWNT-$(CF_2)_3CF_3$ indicates the substituent effect of the fluorine atoms. In contrast, the difference in ΔE between SWNT-$(CH_2)_3CH_3$ and SWNT>$(CH_2)_4$ indicates the effect of the number of reactive sites. Therefore, the similar trends in the difference in ΔE between SWNT-$(CF_2)_3CF_3$ and SWNT>$(CF_2)_4$ and between SWNT-$(CH_2)_3CH_3$ and SWNT>$(CH_2)_4$ indicate that the significant PL wavelength shift in SWNT>$(CF_2)_4$ is caused by the combined effects of cyclisation using a reagent with two reactive sites and the electronic effect of the fluorine atoms. Interestingly, the ΔE of (8,3) SWNT-$(CF_2)_3CF_3$ exhibited different trends from the others. The deviation from the trends of other SWNTs was also observed in the (8,3) SWNT-$(CF_2)_5CF_3$ reported by ref. [21].

**Mechanistic studies on controlling the functionalisation degree and PL characteristics.** To elucidate the origin of the PL selectivity and significant PL wavelength shifts induced by fluorinated alkyl functionalisation, DFT and TD-DFT calculations were

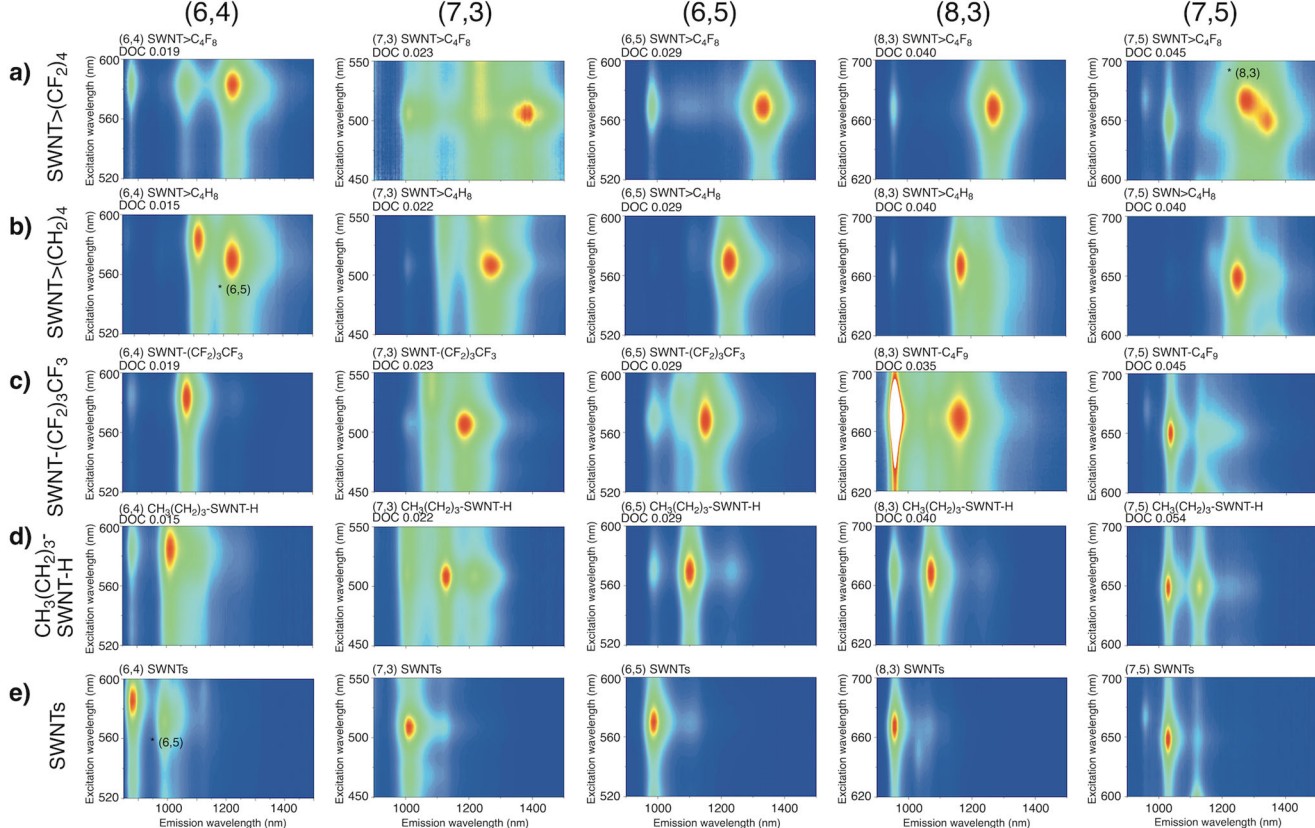

**Fig. 3 Contour plots of the PL intensity as a function of the excitation and emission wavelengths of the separated SWNT adducts in D₂O containing 1 wt% SC. a** SWNT>$(CF_2)_4$. **b** SWNT>$(CH_2)_4$[29]. **c** SWNT-$(CF_2)_3CF_3$. **d** $CH_3(CH_2)_3$-SWNT-H[20]. **e** SWNT.

**Table 1 PL wavelength of the functionalised SWNTs after separation.**

| SWNT | (6,4) | (7,3) | (9,1) | (6,5) | (8,3) | (5,4) | (7,5) |
|---|---|---|---|---|---|---|---|
| SWNT>(CF$_2$)$_4$ | 1064 1222 | 1373 1389 | 1207 | 1320 | 1269 | – | 1345 |
| SWNT>CH$_2$(CF$_2$)$_2$CH$_2$ | 1003 1139 | 1288 | – | 1237 | – | – | – |
| SWNT>(CH$_2$)$_4$[29] | 1115 | 1265 | – | 1229 | 1170 | – | 1248 |
| SWNT-(CF$_2$)$_3$CF$_3$ | 1068 | 1185 | 1153 | 1152 | 1164 | 1027 | 1178 |
| CH$_3$(CH$_2$)$_3$-SWNTs-H[20] | 1015 | 1128 1213 | – | 1104 1231 | 1076 | – | 1136 |
| SWNTs-(CF$_2$)$_5$CF$_3$[21] | 1082 | 1190 | 1128 | 1155 | 1169 | 1027 | 1206 |

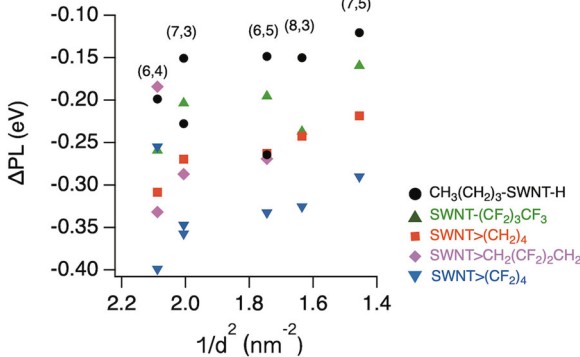

**Fig. 4 Emission energy difference (ΔPL) between the functionalised and non-functionalised SWNTs as a function of the SWNT diameter.** SWNT>(CF$_2$)$_4$ (blue, ▼), SWNT>CH$_2$(CF$_2$)$_2$CH$_2$ (purple, ◆), SWNT>(CH$_2$)$_4$ (red, ■)[29], SWNT-(CF$_2$)$_3$CF$_3$ (green, ▲), and CH$_3$(CH$_2$)$_3$-SWNT-H (black, ●)[20].

performed systematically (Fig. 5). Previous studies[20,22,26,29] have demonstrated that the relative energy of the functionalised SWNTs and the spin density of the intermediates, which depends on the synthesis route and reagents of the functionalised SWNTs, are relevant factors for the binding configuration. Therefore, these factors, namely, spin density of intermediates, relative stability, and transition energy, of SWNT adducts were focused on in this study.

First, to clarify the low functionalisation degree of SWNT adducts using iodofluoroalkanes, as compared to that using 1-iodobutane, the spin density of the fluoroalkyl radicals and monofunctionalised SWNTs radicals were calculated using the UB3LYP method[33−36]. The six configurations of the two carbon atoms in the same hexagonal ring on the SWNTs (Fig. 5c) were classified according to their orientations with respect to the SWNT axis, denoted as L$_{++}$, L$_+$, and L$_-$[23]. As shown in Table 2, the spin densities of monofunctionalised SWNT radicals on SWNTs tended to decrease as the number of fluorine atoms increased. The spin densities of 1,2-L$_{++}$ and 1,2-L$_-$, followed by 1,4-L$_+$ were higher than those of the other carbon atoms. The spin density of the carbon atom in the CF$_3$(CF$_2$)$_3$ radical (0.7880) was lower than that in the CF$_3$(CF$_2$)$_n$(CH$_2$)$_{3-n}$ radicals ($n = 2$ (1.0756), 1 (1.1109), and 0 (1.1104)) and CH$_3$(CH$_2$)$_3$ radical (1.1054)) (Supplementary Table 3). These low spin densities of the carbon atom at the CF$_3$(CF$_2$)$_3$ radicals and the delocalised spin density on monofunctionalised SWNTs are likely responsible for the low functionalisation degree of SWNTs adducts using iodofluoroalkanes.

Next, the transition energy and relative stability (ΔE) of the model compounds of the functionalised SWNTs (Fig. 5) were examined and the results are listed in Table 3. The partial structures of the model compounds are shown in Supplementary

Figs. 19–25. The calculated transition energies showed that the binding configuration is the main factor governing the local band gap of the functionalised SWNTs. Additionally, these transition energies decreased as the number of fluorine atoms in the alkyl groups increased, regardless of the binding configuration, supporting the experimental results where the PL wavelength shifted to longer wavelengths with an increasing number of fluorine atoms.

Focusing on the highest occupied molecular orbital (HOMO) and lowest unoccupied molecular orbital (LUMO) levels, the lower LUMO level of (6,5) SWNT>(CF$_2$)$_4$ than that of (6,5) SWNT>(CH$_2$)$_4$ is a typical effect of the electron-withdrawing groups (Supplementary Figs. 26–28 and Supplementary Table 4). In contrast, the orbital shape or distribution of the HOMO and LUMO does not show significant difference between these two adducts (Supplementary Fig. 29). Figure 5e shows a gradual shift in the transition energy depending on the number of fluorine atoms, which is in good agreement with the PL observations. The transition energy and relative stability for the (6,4), (7,3), (8,3), and (7,5) SWNTs were also examined, and the results are summarised in Supplementary Fig. 30 and Supplementary Tables 5–7. Most of the SWNTs adducts, except for the L$_+$ (6,4) and (8,3) SWNT adducts, exhibited a similar trend to that of the (6,5) SWNTs and agreed well with the experimental measurements; that is, the substitution of hydrogen atoms with fluorine atoms was effective in decreasing the local band gap.

A comparison of the relative stability tendency between the positional isomers of each functionalised SWNT showed that the substituted fluorine atoms influenced the relative stability, but the order of the binding configurations was similar to that of the corresponding alkylated SWNTs. The relative stability of the 1,2-adducts of (6,5) SWNT-[(CF$_2$)$_3$CF$_3$]$_2$ was significantly lower than that of the corresponding 1,4-adducts. The energy difference between the 1,2- and 1,4-adducts, as compared to that between the corresponding alkylated SWNTs, (6,5) SWNT-[(CH$_2$)$_3$CH$_3$]$_2$, was very large, suggesting a larger steric hindrance between the perfluoroalkyl groups. In addition to the low spin density of the intermediates, it is possible that steric hindrance suppressed bis-perfluoroalkylation, resulting in selective hydroperfluoroalkylation and the emergence of a single emission from SWNT-(CF$_2$)$_3$CF$_3$ at 1152 nm.

As mentioned above, the spin density of plausible intermediates and the relative stability of functionalised SWNTs exhibited similar trends in the corresponding perfluoroalkylation and alkylation reactions, suggesting that the binding configurations of functionalised SWNTs using SWNTs-R are the same. Consequently, the theoretical calculation results indicate that the origin of the significantly longer wavelength-shifted PL peak in SWNT>(CF$_2$)$_4$ is due to the substituent effect of the fluorine atom and the molecular design of the reactant with two reaction sites. Previous studies proposed that SWNT-(CH$_2$)$_3$CH$_3$ may obtain the thermodynamic product[20], that is,

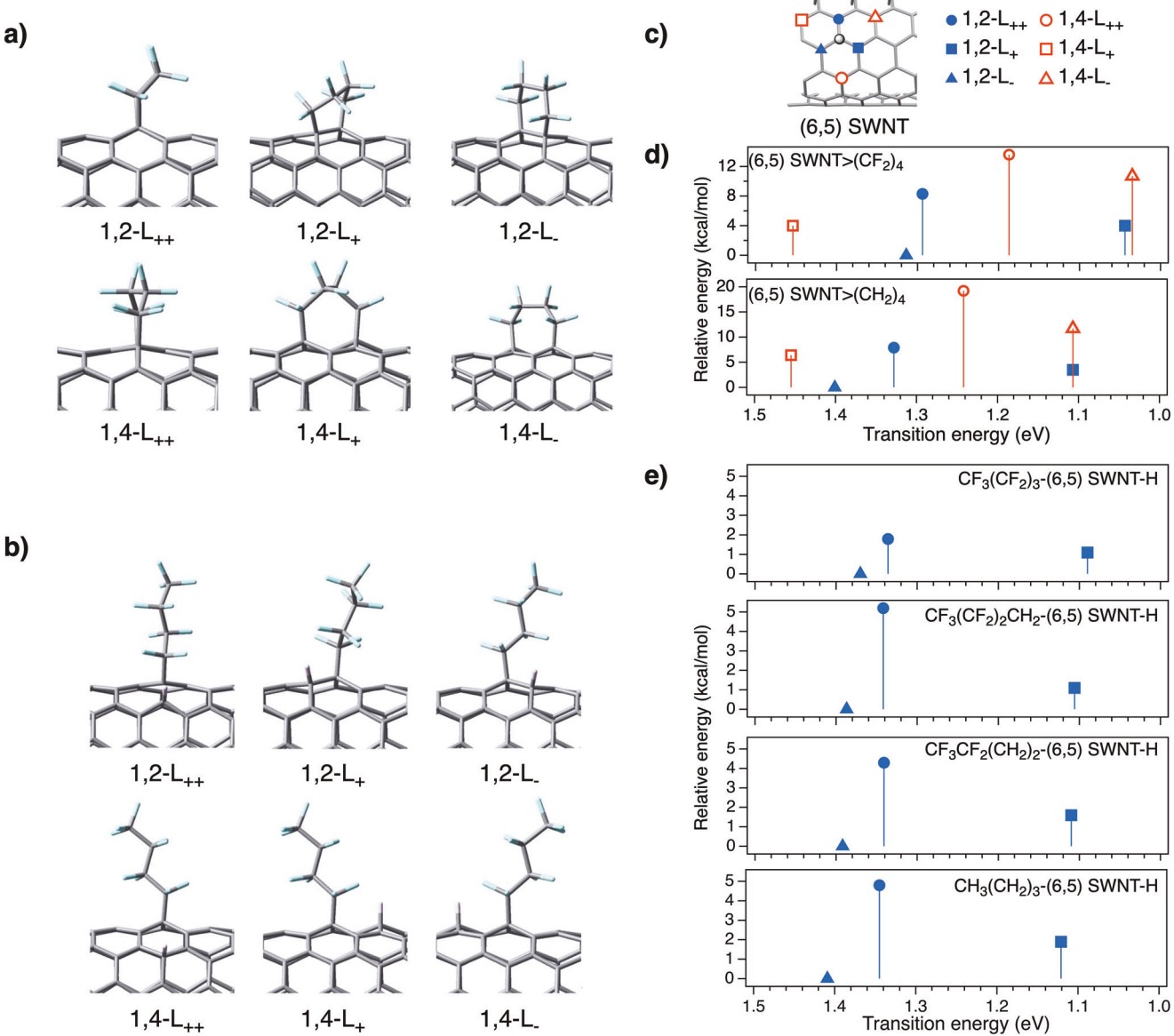

**Fig. 5 Optimised structures, transition energies, and relative energies of SWNT adducts.** Optimised partial structures of **a** (6,5) SWNT>(CF$_2$)$_4$ and **b** CF$_3$(CF$_2$)$_3$-(6,5) SWNT-H. **c** Six binding configurations of (6,5) SWNTs. The binding sites relative to the carbon atoms highlighted in the grey circle are presented as 1,2-L$_{++}$ (●), 1,2-L$_{+}$ (■), 1,2-L$_{-}$ (▲), 1,4-L$_{++}$ (○), 1,4-L$_{+}$ (□), and 1,4-L$_{-}$ (△). **d, e** Calculated transition energies (eV) and relative energies (kcal/mol) of the model molecules of (6,5) SWNT>(CF$_2$)$_4$, (6,5) SWNT>(CH$_2$)$_4$, CF$_3$(CF$_2$)$_3$-(6,5) SWNT-H, CF$_3$(CF$_2$)$_2$CH$_2$-(6,5) SWNT-H, CF$_3$CF$_2$(CH$_2$)$_2$-(6,5) SWNT-H, and CH$_3$(CH$_2$)$_3$-(6,5) SWNT-H.

**Table 2 Spin densities of the 1,2-L$_{++}$, 1,2-L$_{+}$, 1,2-L$_{-}$, 1,4-L$_{++}$, 1,4-L$_{+}$, and 1,4-L$_{-}$ sites for (6,5) SWNT-(CF$_2$)$_3$CF$_3$, -CH$_2$(CF$_2$)$_2$CF$_3$, -(CH$_2$)$_2$CF$_2$CF$_3$, -(CH$_2$)$_3$CF$_3$, -(CH$_2$)$_3$CH$_3$ radicals at the UB3LYP/3-21 G level of theory based on the 3-unit (6,5) SWNT model.**

| | (6,5) SWNT-(CF$_2$)$_3$CF$_3$ radical | (6,5) SWNT-CH$_2$(CF$_2$)$_2$CF$_3$ radical | (6,5) SWNT-(CH$_2$)$_2$CF$_2$CF$_3$ radical | (6,5) SWNT-(CH$_2$)$_3$CF$_3$ radical | (6,5) SWNT-(CH$_2$)$_3$CH$_3$ radical |
|---|---|---|---|---|---|
| 1,2-L$_{++}$ | 0.2198 | 0.2230 | 0.2242 | 0.2259 | 0.2281 |
| 1,2-L$_{+}$ | 0.1538 | 0.1635 | 0.1575 | 0.1583 | 0.1590 |
| 1,2-L$_{-}$ | 0.2163 | 0.2265 | 0.2210 | 0.2214 | 0.2215 |
| 1,4-L$_{++}$ | 0.1595 | 0.1703 | 0.1699 | 0.1701 | 0.1701 |
| 1,4-L$_{+}$ | 0.1925 | 0.1915 | 0.1953 | 0.1955 | 0.1961 |
| 1,4-L$_{-}$ | 0.1280 | 0.1290 | 0.1297 | 0.1305 | 0.1318 |

The position of sp$^2$ carbon atoms at the 1,2- and 1,4-positions of the sp$^3$ carbon atom is denoted as 1,2-L$_{87}$, 1,2-L$_{27}$, 1,2-L$_{-33}$, 1,4-L$_{87}$, 1,4-L$_{27}$, and 1,4-L$_{-33}$. The subscripts of L denote the positive or negative angles of the sp$^2$ and sp$^3$ carbon atoms on SWNTs relative to the axis.

**Table 3 Transition energies (in nm) and relative energies ($\Delta E$, in kcal/mol) of SWNT adducts.**

| Binding configuration | 1,2-L$_{++}$ | 1,2-L$_+$ | 1,2-L$_-$ | 1,4-L$_{++}$ | 1,4-L$_+$ | 1,4-L$_-$ |
|---|---|---|---|---|---|---|
| SWNTs | Transition energy (nm) | | | | | |
| (6,5) SWNT>(CF$_2$)$_4$ | 958 | 1188 | 944 | 1045 | 853 | 1198 |
| (6,5) SWNT>(CH$_2$)$_4$[29] | 934 | 1120 | 885 | 998 | 852 | 1120 |
| CF$_3$(CF$_2$)$_3$-(6,5) SWNT-H | 928 | 1138 | 905 | 1009 | 876 | 1138 |
| CH$_3$(CH$_2$)$_3$-(6,5) SWNT-H[20] | 922 | 1106 | 880 | 992 | 855 | 1121 |
| | Relative energy ($\Delta E$, kcal/mol) | | | | | |
| (6,5) SWNT>(CF$_2$)$_4$ | 8.3 | 4.0 | 0 | 13.6 | 4.0 | 10.7 |
| (6,5) SWNT>(CH$_2$)$_4$[29] | 7.9 | 3.5 | 0 | 18.2 | 6.4 | 11.7 |
| CF$_3$(CF$_2$)$_3$-(6,5) SWNT-H | 1.8 | 1.1 | 0 | 4.6 | 5.0 | 8.3 |
| CH$_3$(CH$_2$)$_3$-(6,5) SWNT-H[20] | 4.8 | 1.9 | 0 | 6.1 | 6.6 | 10.3 |

the 1,2-addition product (1,2-L$_-$ for mod($n-m$,3) = 1, 1,2-L$_+$ for mod($n-m$,3) = 2), whereas SWNT>(CH$_2$)$_4$ may result in the kinetic product, that is, the 1,2-L$_{++}$-addition product[29]. These findings suggest that SWNT-(CF$_2$)$_3$CF$_3$ prefers the 1,2-L$_-$ for mod($n-m$,3) = 1 and the 1,2-L$_+$ for mod($n-m$,3) = 2, while SWNT>(CF$_2$)$_4$ is more likely to produce the 1,2-L$_{++}$ adducts, respectively. Accordingly, the calculated transition energies of these adducts exhibit a trend similar to that observed in the PL wavelength shifts. The effect of the tether length of two binding sites in reactive alkylation using 1,n-dibromoalkane (Br(CH$_2$)$_n$Br) has been reported previously, and the results indicate that the alkylation reagents having the appropriate tether length ($n = 3$–5) induce a new single PL peak in SWNTs[29]. It is known that the small ring-forming reaction is stereoelectronically favourable and has high reaction efficiency. The higher PL selectivity observed in the reaction using I(CF$_2$)$_4$I and ICH$_2$(CF$_2$)$_4$CH$_2$I is due to their suitable tether lengths resulting in higher efficiency of the cycloaddition reaction.

In conclusion, SWNTs were functionalised using 1-iodofluorobutane derivatives, and the results revealed that the number of fluorine atoms was effective in controlling the degree of functionalisation and selectivity of the binding configuration. Additionally, the PL peak selectivity emerged at significantly longer wavelengths by the functionalisation of I(CF$_2$)$_4$I, as compared with the corresponding alkylation. The PL wavelength of (6,5) SWNT>(CF$_2$)$_4$ was observed at 1320 nm, which is suitable for telecommunication applications. The results revealed that (6,5) SWNTs can selectively produce new NIR PL in the range of 1100–1320 nm, which is largely red-shifted from the pristine PL wavelength at 970 nm, by modifying the number of reactive sites and fluorine atoms in the alkylation reagents. In addition to the optical resolution of (11,−5) and (6,5) SWNTs, highly pure (6,4) and (8,3) SWNT>(CF$_2$)$_4$ were successfully obtained using gel chromatography. The PL wavelengths of (6,4), (7,3), (8,3), and (7,5) SWNT>(CF$_2$)$_4$ and SWNT-(CF$_2$)$_3$CF$_3$ were observed at longer wavelengths than those of their corresponding SWNT>(CH$_2$)$_4$[29] and SWNT-(CH$_2$)$_3$CH$_3$ adducts[20]. These results indicate the effectiveness of the functionalisation using perfluoroalkanes with one or two reactive sites for the large band gap modulation. The PL wavelength differences before and after functionalisation exhibited a diameter dependence. The new findings in this study increase our understanding of the functionalised SWNT structure and methods for controlling the local band gap, which will contribute to the optical application of SWNTs, such as in NIR light-emitting materials.

## Methods

**Materials**. The materials were purchased and used without further purification unless otherwise stated. Tetrahydrofuran (pure, anhydrous) was obtained from Kanto Chemical and purified using a Glass Contour Ultimate solvent purification system (Nikko Hansen Co., Ltd.). Ethanol was distilled over magnesium. 2,2,3,3-Tetrafluoro-1,4-diiodobutane was prepared according to the reported procedure, described in the Supporting Information, Supplementary Figs. 31–36, and supplementary Methods.

**Functionalisation of SWNTs**. Naphthalene (300 mg, 2.34 mmol) and sodium (156 mg, 6.80 mmol) were placed in a 200 mL heat-dried three-necked round-bottom flask under argon. Anhydrous tetrahydrofuran (100 mL) was added to the flask, and the mixture was stirred for 1 h. A portion of SWNTs (10 mg, SG65i, Sigma-Aldrich) in a 200 mL heat-dried three-necked round-bottom flask was placed under argon. Sodium naphthalenide solution was added to the SWNTs, which were then sonicated for 1 h, followed by the addition of 1,4-diiodoocta-fluorobutane (0.260 mL, 1.405 mmol) to the mixture. The mixture was sonicated for 30 min and quenched by the addition of 4 mL dry ethanol (Kanto Chemical). The resulting black suspension was filtered using a PTFE filter (0.1 μm) and washed with tetrahydrofuran, acetone, methanol, and water using a dispersion-filtration process. The solid was then dried under vacuum.

**Separation of functionalised SWNTs**. Chiral index dependent separation was performed using gel chromatography. Approximately 2.0 mg of SWNTs were added to a vial containing 2.5 mL of aqueous 1.0 wt% SC (≥98%). The vial was bath-sonicated for 3 h (Branson; Bransonic Ultrasonic Cleaner 2510J-MT). The resulting dispersion was centrifuged at 140,000 × $g$ for 1 h in a high-speed centrifuge equipped with an S58A angle rotor (Koki Holdings Co., Ltd.; micro ultracentrifuge CS 100FNX). 2.0 mL of aqueous 2.0 wt% SDS (≥97%) was added to the supernatant solution before the HPLC separation. The helicity sorting and optical resolution of the SWNTs were achieved via HPLC using a JASCO ChromNAV system equipped with a JASCO LC-Net II interface, a JASCO PU-2089i gradient inert pump, a JASCO MD-4010 photo diode array detector, a CO-4060 column oven (23 °C), and a column ($\phi$10 × 200 mm) filled with the gel (Sephacryl S-200, Cytiva). The flow rate was 2.0 mL min$^{-1}$ and the sample (SWNTs dispersion) injection volume was 3 mL. The eluent was an aqueous solution containing 0.5 wt% SC + 0.5 wt% SDS + X wt% DOC (>96%). The concentration of DOC (X) increased from 0 to 1. Fractions were collected every 5 mL using a fraction collector.

**Optical measurements**. Optical absorption spectra were recorded using a spectrophotometer (V-670; JASCO Corp.), using a quartz cell with a path length of 10 mm (Supplementary Table 1). Raman spectra ($\lambda_{ex} = 561$ nm) were recorded using a spectrophotometer (LabRAM HR-800; HORIBA Ltd.) (Supplementary Table 1). These spectra were normalised relative to the G-band. The NIR fluorescence intensity as a function of the excitation and emission wavelengths of the dispersed SWNTs was obtained using a spectrophotometer equipped with a 450 W lamp and a Symphony-II CCD detector (Nanolog; HORIBA Ltd.). The excitation wavelength was varied from 400–1000 or 1100 nm in steps of 1 nm. The excitation and emission spectral slit widths were both set to 10 nm. The PL intensity was normalised by the integration time. PL spectra were recorded at a 90° angle relative to the excitation light unless otherwise stated. The CD spectra of the separated samples were measured at 20 °C using a CD spectropolarimeter (J-820; JASCO Corp.).

**Computational details**. One unit-cell of (6,4), (7,3), (6,5), (8,3), and (7,5) SWNT was adopted as the computational model for the functionalised SWNTs (Supplementary Data 2). The dangling bonds at the terminals were passivated with hydrogen atoms to eliminate midgap trap states. Two functional groups were attached at the 1,2- and 1,4-positions in the same hexagonal ring. The six different binding configurations of the two substituents were classified according to their orientations with respect to the SWNT axis, denoted as L$_{++}$, L$_+$, and L$_-$. Geometric optimisations of the functionalised SWNTs were performed using DFT at the B3LYP[33–35]/6–31 G* level of theory[36–38] (Supplementary Data 2). Partial structures are shown in Supplementary Figs. 19–25. TD-DFT calculations were performed at the B3LYP/3-21 G level of theory[39]. The spin densities on the six binding configurations for the n-butylated (6,5) SWNT radical were obtained using a

computational model consisting of 3-unit cells, which were calculated at the UB3LYP/3-21 G level of theory. The spin densities of the $CF_3(CF_2)_3$, $CF_3(CF_2)_2CH_2$, $CF_3CF_2(CH_2)_2$, $CF_3(CH_2)_3$, and $CH_3(CH_2)_3$ radicals were also calculated at the UB3LYP/3-21 G level of theory. All DFT calculations were performed using the Gaussian 09 software suite, version E.01[40].

## Data availability

Synthetic and characterization data for all reported new compounds as well as computational details are included in the Supplementary Information file. Text data for Figs. 1b, d, and 2 are included in Supplementary Data 1.

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

## Acknowledgements

This study was supported by JSPS KAKENHI Grant-in-Aid for Scientific Research (B) (21H01759, 20H02210, 20H02718, 17H02735) and Transformative Research Areas (A) (22H05133). This study was performed under the Cooperative Research Programme of "Network Joint Research Centre for Materials and Devices." We thank Dr. Mitsuaki Suzuki of Josai University for the MS spectrometry analysis.

## Author contributions

Y.M. conceived the idea of this study and designed the experiments. Nanotube functionalisation, separation, and optical characterisation were performed by Y.S., N.K., and Y.K. XPS analysis was conducted by A.D. H.K. and M.M. Theoretical simulations were performed by P.Z. and M.E. All authors contributed to the analysis and interpretation of results. Y.M., M.Y., and M.E. wrote the manuscript with the assistance of all co-authors.

## Competing interests

The authors declare no competing interests.
