## [Peer Review File · Communications Chemistry]

Reviewers' comments:

Reviewer #1 (Remarks to the Author):

In this work by Maeda et al., the authors enhanced the optical properties of (6,5) SWCNTs by using functionalization with perfluoroalkylated moieties. The results are interesting and nicely correspond with the aims and scope of the journal. However, certain issues should be addressed before the paper can be reconsidered for publication. Please refer to the suggestions given below:

- 1) The SWNT-(CF₂)₄ notation is misleading as it suggests the grafting of linear (CF₂)₄ chain onto the SWNT surface. Please consider if SWNT>(CF₂)₄ is more appropriate. If you agree, the notation on other samples should be changed analogously as well.
- 2) Why the spectral lineshapes are, to a large degree, lost in many plots in Fig. 1a? How do they correlate with the reported reactivity differences?
- 3) "The spectral changes after functionalisation tended to decrease with an increasing the number of fluorine atom substitutions (Supplementary Table 1)" - is this conclusion coherent with the Raman results shown in Fig. S1? SWNT-(CF₂)₄ are much more disordered than SWNT-CH₂(CF₂)₂CH₂ even though four CF₂ groups were introduced not two. Besides that, there are Raman spectra of 7 samples (Figs. S1 and S2), whereas 9 samples (Fig. 1) were characterized by absorption and PL. Please supplement the missing data.
- 4) Not all plots from SI are referenced and discussed in the main text.
- 5) Do you mean "Conclusions" instead of "Discussion" (Line 247)?

Reviewer #2 (Remarks to the Author):

This manuscript describes the functionalization of single-walled carbon nanotubes (SWNTs) via cyclic perfluoroalkylation. Various 1-iodobutane derivatives were used in conjunction with 1,4-diiodo-2,2,3,3-tetrafluorobutane and I(CF₂)₄I to investigate the effects of fluorine atoms and number of reactive sites (one or two) on the functionalization. Absorbance spectra and photoluminescence excitation (PLE) maps were used to characterize the degree of functionalization of the predominantly (6,5) samples through the emergence of two new NIR photoluminescence (PL) peaks, denoted E11* and E11**. Gel chromatography was subsequently performed to separate other chiralities from the sample and investigate the effects of the functionalization. In general, it was found that the degree of functionalization decreased with the number of fluorine atoms and increased with the number of reactive sites; however, the PL wavelength shift of E11* and E11** was larger for the fluorinated molecules. Notably, a highly fluorinated molecule with two reactive sites shifts the PL wavelength to the telecom region of >1300 nm for (6,5) SWNTs.

General Comments and Recommendation

This work builds off previous SWNT functionalization studies by Maeda et. al. It is evident that a lot of effort went into performing this detailed and systematic study, and the general scientific area of the work would be of interest to those in the field of nanotube functionalization and photophysics. While there are some issues/clarifications/inconsistencies that should be addressed, the main concern is the

overall nomenclature used throughout the manuscript that renders it challenging to follow. It is recommended that the manuscript be published in Communications Chemistry pending significant editorial modification to present the data and results more clearly.

Specific Comments

1. Editorial comments

- This manuscript provides a systematic study of the effects of fluorine atoms and the number of reactive sites. Therefore, the chemical formulas for the 8 derivatives are all very similar with only a single atom differentiating them. It would be helpful to provide a figure with each of the different substituents, perhaps labeling the part of the molecule that is being systematically changed, and give each molecule a simplified name. For example, the authors published a similar study where they used simplified names rather than the chemical formula to represent the different derivatives (RSC Adv., 2019, 9, 13998–14003). Additionally, Wang et. al. published a systematic study of functionalization where the property being changed (number of fluorine atoms, length of chain) is highlighted (J. Am. Chem. Soc. 2016, 138, 6878–6885). Both methods are effective in helping the reader easily follow the logic behind the study and the results.

- There are several awkward phrases in the manuscript that should be addressed.

2. Figures

- The figures in the manuscript emphasize the amount of work that went into this study. Because of this, some of the figures are very busy, and in conjunction with the nomenclature described above, can make it challenging to follow. Modifications to the nomenclature would certainly help with this, but additional changes to the figures would also prove beneficial. For example, the matrix in Figure 3 shows the effects of the functionalization on various (n,m) chiralities. Perhaps labeling each column with the respective chirality, and each row with the respective derivative, would help highlight the observed trends. Additionally, in Figure 1, perhaps having the absorbance spectrum with the corresponding PLE map side-by-side and ordered by the systematic change would help highlight the observed trend.

- In Figure S7, each plot has three spectra representing the same derivative. Are these three different trials of the same functionalization?

3. Discussion

- It appears functionalization causes the emergence of the new PL peaks, E11* and/or E11**. However, the reason for why one or both peaks occur with specific derivatives and/or chiralities was not discussed. Why do one (or both) of the peaks emerge as a result of the specific functionalization, and how does that relate to the chirality?

- There was discussion regarding why fluorine derivatives are less reactive and why they result in PL at longer wavelengths, but why is having 2 binding sites more efficient and more selective?

- It was mentioned that “Br(CH₂)₄Br and I(CF₂)₄I, with two reactive sites, exhibited higher reactivity than CH₃(CH₂)₃I and CF₃(CF₂)₃I.” However, there was no further discussion on the bromine comparison.

- CD was performed on the separated fractions and seem to match results from a previous manuscript by the authors. However, there is no discussion on the relevance of this measurement with respect to this manuscript, and it is therefore unclear how CD adds to the interpretation of the results.

- Is there a reason SWNT-CH₂(CF₂)₂CF₃ was not separated using gel chromatography, as the derivatives used for separation were chosen to look at chiralities other than (6,5) and this derivative has the largest amount of (6,4).

4. Clarifications/inconsistencies

- Page 6: there seems to be some inconsistencies with the derivatives presented in the manuscript. Specifically, SWNT-(CH₂)₄ is discussed throughout the results but is not represented in Table S1 nor included in the list of functionalized SWNTs that are being studied.

- Page 6: it should be clarified that “the spectral changes after functionalization tended to decrease with an increasing number of fluorine atom substitutions” was concluded from the D/G ratio.

- Page 8: It is stated that “... gel chromatography was conducted on SWNT-(CF₂)₄, SWNT-CH₂(CF₂)₂CH₂, and SWNT-(CF₂)₃CF₃”. In Figure 4 and Table 1, which shows the results of the separated SWNTs, SWNT-(CH₂)₄ and SWNT-CH₃(CH₂)₃-SWNT-H (derivatives not listed above) are both shown, but SWNT-CH₂(CF₂)₂CH₂ is not. The derivatives shown in the figures/tables should be consistent with what is described in the manuscript.

- Page 8: It is stated that “... CF₃(CF₂)₂CH₂I, CF₃(CF₂)₃I, and I(CF₂)₄I produced a single E11* or E11** PL peak, whereas the other reagents produced two PL peaks.” However, on page 7 it is stated that “... one prominent PL peak attributed to the functionalised (6,5) SWNTs was observed from SWNT-(CF₂)₃CF₃ (1152 nm), SWNT-CH₂(CF₂)₂CF₃ (1119 nm), SWNT-(CH₂)₄ (1230 nm), SWNT-CH₂(CF₂)₂CH₂ (1246 nm), and SWNT-(CF₂)₄ (1318 nm).” Should SWNT-(CH₂)₄ and SWNT-CH₂(CF₂)₂CH₂ be included on page 8? If not, an explanation as to why they are an exception should be provided.

- Page 10: It appears (7,3) SWNT-(CF₂)₄ also exhibited two peaks in the PLE map addition to (6,4). If it should not be included, an explanation as to why should be provided.

- Page 10: The term “high PL wavelength selectivity” is somewhat confusing. Do the authors mean the derivative favors longer PL wavelengths, or there is high selectivity in the functionalization?

- Page 19: what is meant by “normalized to the measurement time”?

Reviewer #3 (Remarks to the Author):

I have reviewed "Selective Emergence of Photoluminescence at Telecommunication Wavelengths from Cyclic Perfluoroalkylated Carbon Nanotubes" by Maeda, et. Al. I believe the results presented are interesting and novel. However, they are iterative on previous efforts to produce SWCNT functionalization that emits in the NIR range. The authors successfully produce a functionalization scheme that generates emission energies of wavelengths longer than 1300 nm. The methods used, however, are a slight variation on previous functionalization schemes that seem to introduce additional electron inducing effects as opposed to generating different configurations or defect isomers. As such, this article is of interest to specialists in the field and has, in my opinion, little general relevance. I think such a report would be well catered for a publication that focuses on nanotechnology or synthetic methods and would not recommend publication in Communications Chemistry.

Point-by-point response to the referees

Reviewer #1 (Remarks to the Author):

In this work by Maeda et al., the authors enhanced the optical properties of (6,5) SWCNTs by using functionalization with perfluoroalkylated moieties. The results are interesting and nicely correspond with the aims and scope of the journal. However, certain issues should be addressed before the paper can be reconsidered for publication. Please refer to the suggestions given below:

We greatly appreciate your helpful comments.

The manuscript has been modified according to your comments.

Revisions are shown with red characters.

1) The SWNT-(CF₂)₄ notation is misleading as it suggests the grafting of linear (CF₂)₄ chain onto the SWNT surface. Please consider if SWNT>(CF₂)₄ is more appropriate. If you agree, the notation on other samples should be changed analogously as well.

We agree with the suggestion. The SWNTs-(CF₂)₄ and SWNTs-(CH₂)₄ notations were revised to SWNTs>(CF₂)₄ and SWNTs>(CH₂)₄, respectively, throughout the manuscript and supplementary information.

2) Why the spectral lineshapes are, to a large degree, lost in many plots in Fig. 1a? How do they correlate with the reported reactivity differences?

We interpret that the characteristic absorption is reduced due to the high functionalization degree in Fig. 1. Previously, we reported the steric effect of bromoalkanes on the functionalization of SWNTs. The characteristic absorption peaks decreased significantly by the reaction using 1-bromopropane and 1-bromobutane. Compared to these results, the spectral changes in absorption spectra were suppressed by the reaction using the sterically hindered 2-bromopropane and 2-bromo-2-methylpropane. The significant decrease of absorption peak intensity observed in SWNT-(CH₂)₃CH₃ prepared with 1-iodobutane might be due to the higher reactivity of iodoalkane than bromoalkane. The D/G values of SWNT adducts are shown in Supplementary Table 1.

Figure. Absorption of SWNTs and functionalised SWNTs prepared using 1-bromopropane, 2-bromopropane, 1-bromobutane, and 2-bromo-2-methylpropane. (*Chem. Eur. J.* **2023**, e202300766.)

3) "The spectral changes after functionalisation tended to decrease with an increasing the number of fluorine atom substitutions (Supplementary Table 1)" - is this conclusion coherent with the Raman results shown in Fig. S1? SWNT-(CF₂)₄ are much more disordered than SWNT-CH₂(CF₂)₂CH₂ even though four CF₂ groups were introduced not two. Besides that, there are Raman spectra of 7 samples (Figs. S1 and S2), whereas 9 samples (Fig. 1) were characterized by absorption and PL. Please supplement the missing data.

The amount of the spectral changing after the functionalisation was evaluated from the D/G values of the Raman spectra, as shown in Supplementary Figure 1. As the reviewer pointed out, the correlation between the amount of spectral changes and the number of fluorine atoms in SWNT-CH₂(CF₂)₂CH₂ and SWNT-CH₂(CF₂)₂CF₃ is not sufficient. Therefore, we revised the manuscript. We added Raman, absorption, and PL spectra of SWNT-(CH₂)₄ in Supplementary Figs. 1 and 2. Raman spectra of SWNTs was shown with each SWNT adducts. The sentence was revised as follow.

Revised P.6

‘With the exception of SWNT-CH₂(CF₂)₂CF₃ and SWNT-(CF₂)₂CH₂, the D/G values after functionalisation tended to decrease with an increasing number of fluorine atoms (Supplementary Table 1).’

4) Not all plots from SI are referenced and discussed in the main text.

The Supplementary Tables and Figures were referenced and discussed in the main text.

5) Do you mean "Conclusions" instead of "Discussion" (Line 247)?

We revised the Results to Results and Discussion.

Reviewer #2 (Remarks to the Author):

This manuscript describes the functionalization of single-walled carbon nanotubes (SWNTs) via cyclic perfluoroalkylation. Various 1-iodobutane derivatives were used in conjunction with 1,4-diiodo-2,2,3,3-tetrafluorobutane and I(CF₂)₄I to investigate the effects of fluorine atoms and number of reactive sites (one or two) on the functionalization. Absorbance spectra and photoluminescence excitation (PLE) maps were used to characterize the degree of functionalization of the predominantly (6,5) samples through the emergence of two new NIR photoluminescence (PL) peaks, denoted E11 and E11**. Gel chromatography was subsequently performed to separate other chiralities from the sample and investigate the effects of the functionalization. In general, it was found that the degree of functionalization decreased with the number of fluorine atoms and increased with the number of reactive sites; however, the PL wavelength shift of E11* and E11** was larger for the fluorinated molecules. Notably, a highly fluorinated molecule with two reactive sites shifts the PL wavelength to the telecom region of >1300 nm for (6,5) SWNTs.*

General Comments and Recommendation

This work builds off previous SWNT functionalization studies by Maeda et. al. It is evident that a lot of effort went into performing this detailed and systematic study, and the general scientific area of the work would be of interest to those in the field of nanotube functionalization and photophysics. While there are some issues/clarifications/inconsistencies that should be addressed, the main concern is the overall nomenclature used throughout the manuscript that renders it challenging to follow. It is recommended that the manuscript be published in Communications Chemistry pending significant editorial modification to present the data and results more clearly.

We greatly appreciate your helpful comments.

The manuscript has been modified according to your comments.

Revisions are shown with red characters.

Specific Comments

1. Editorial comments

- This manuscript provides a systematic study of the effects of fluorine atoms and the number of reactive sites. Therefore, the chemical formulas for the 8 derivatives are all very similar with only a single atom differentiating them. It would be helpful to provide a figure with each of the different substituents, perhaps labeling the part of the molecule that is being systematically changed, and give each molecule a simplified name. For example, the authors published a similar study where they used simplified names rather than the chemical formula to represent the different derivatives (RSC Adv., 2019, 9, 13998–14003). Additionally, Wang et. al. published a systematic study of functionalization where the property being changed (number of fluorine atoms, length of chain) is highlighted (J. Am. Chem. Soc. 2016, 138, 6878–6885). Both methods are effective in helping the reader easily follow the logic behind the study and the results.

- There are several awkward phrases in the manuscript that should be addressed.

According to the reviewer's suggestion, we provide a list of reagents and SWNT adducts highlighted fluoroalkane groups in red, as shown in Fig. 1.

Revised Fig. 1a.

a) Reagent	SWNT adduct
I-(CF ₂) ₄ -I	SWNT>(CF ₂) ₄
I-CH ₂ (CF ₂) ₂ CH ₂ -I	SWNT>CH ₂ (CF ₂) ₂ CH ₂
Br-(CH ₂) ₄ -Br	SWNT>(CH ₂) ₄
I-(CF ₂) ₃ CF ₃	SWNT-(CF ₂) ₃ CF ₃
I-CH ₂ (CF ₂) ₂ CF ₃	SWNT-CH ₂ (CF ₂) ₂ CF ₃
I-(CH ₂) ₂ CF ₂ CF ₃	SWNT-(CH ₂) ₂ CF ₂ CF ₃
I-(CH ₂) ₃ CF ₃	SWNT-(CH ₂) ₃ CF ₃
I-(CH ₂) ₃ CH ₃	SWNT-(CH ₂) ₃ CH ₃

2. Figures

- The figures in the manuscript emphasize the amount of work that went into this study. Because of this, some of the figures are very busy, and in conjunction with the nomenclature described above, can make it challenging to follow. Modifications to the nomenclature would certainly help with this, but additional changes to the figures would also prove beneficial. For example, the matrix in Figure 3 shows the effects of the functionalization on various (n,m) chiralities. Perhaps labeling each column with the respective chirality, and each row with the respective derivative, would help highlight the observed trends. Additionally, in Figure 1, perhaps having the absorbance spectrum with the corresponding PLE map side-by-side and ordered by the systematic change would help highlight the observed trend.

The absorption spectra with the corresponding PLE map were displayed side-by-side in Figure 1. The chiral index and notation of SWNT adducts were added in Fig.3.

Revised Figure 1

Figure 1. Absorption and PL spectra of functionalized SWNT adducts. **a** Chemical formulae of the reagents and abbreviations of the corresponding SWNT adducts. **b,d** Absorption spectra and **c,e** PL mapping of SWNTs and functionalised SWNTs with increasing numbers of fluorine substituents dispersed in D_2O containing 1 wt% SDBS. The peak assignment for (6,5) SWNT (E_{22} and E_{11} absorption, E_{11} , E_{11}^* , and E_{11}^{**} PL) is shown unless otherwise stated.

Revised Figure 3

Figure 3. Contour plots of the PL intensity as a function of the excitation and emission wavelengths of the separated SWNT adducts in D₂O containing 1 wt% SC. CH₃(CH₂)₃-SWNT-H²⁰. SWNT-(CF₂)₃CF₃. SWNT-(CH₂)₄²⁹. SWNT-(CF₂)₄.

- In Figure S7, each plot has three spectra representing the same derivative. Are these three different trials of the same functionalization?

There are the results of three different trials of the same functionalisation. To avoid the readers misleading, Fig. S2, S4, and S7 were deleted.

3. Discussion

- It appears functionalization causes the emergence of the new PL peaks, E₁₁* and/or E₁₁**. However, the reason for why one or both peaks occur with specific derivatives and/or chiralities was not discussed. Why do one (or both) of the peaks emerge as a result of the specific functionalization, and how does that relate to the chirality?

The selective emergence of E₁₁* PL in the reaction with fluorine substituted iodoalkane was explained

by the spin density of reaction intermediates and relative energy of the model compound isomers in the manuscript. The low spin density at the reaction sites and steric repulsion of the fluoroalkyl groups compared to the corresponding alkyl groups suppressed dialkylation, resulting in the formation of hydroalkylated adducts, selectively. The control of the two PL peaks by hydroalkylation and dialkylation was reported previously. SWNT reacted with butyllithium and bromobutane emerged E_{11}^{**} PL selectively and SWNT reacted with butyllithium and trifluoroacetic acid emerged E_{11}^* , predominantly.²⁰ The comparison of experimental results with theoretical calculations of the corresponding model compounds indicated that hydroalkylated SWNT adducts and dialkylated SWNT adducts are plausible candidates for generating the E_{11}^* and E_{11}^{**} PL peaks. We revised related sentence as follow.

Revised P.12

‘To elucidate the origin of the **PL selectivity and** significant PL wavelength shifts induced by fluorinated alkyl functionalisation, DFT and TD-DFT calculations were performed systematically (**Figure 5**). Previous studies^{20,22,26,29} have demonstrated that the relative energy of the functionalised SWNTs and the spin density of the intermediates, **which depends on the synthesis route and reagents of the functionalised SWNTs, are relevant factors for the binding configuration.** Therefore, these factors, namely, **spin density of intermediates, relative stability, and transition energy, of SWNT adducts** were focused on in this study’

As reviewer pointed out, the two PL peaks were observed in (6,4) and (7,3) SWNT-(CF₂)₄, and the relation of the chirality is interesting. It is currently unclear why the two peaks are observed in (6,4) and (7,3) SWNT-(CF₂)₄ even in the theoretical calculations. SWNT-(CH₂)₄ showed high selectivity in the emergence of new PL peak regardless of chirality. Since the functionalization degree of SWNT-(CF₂)₄ was lower than that of SWNT-(CH₂)₄, it is possible that substitution with a fluorine atom reduce the reactivity, decreasing the selectivity of the cycloaddition. On the other hand, the reactivity of small diameter SWNT is known to be higher than that of larger ones. Therefore, the consistent reason for the chirality dependent PL selectivity is not clear. For the clarification, it requires further consideration. The sentence was added as follow.

Revised P. 10

‘**Interestingly, two PL peaks emerged in (6,4) and (7,3) SWNT-(CF₂)₄; however, the chirality dependence on the selectivity is currently unclear.**’

- There was discussion regarding why fluorine derivatives are less reactive and why they result in PL at longer wavelengths, but why is having 2 binding sites more

efficient and more selective?

The effect of the tether length of two binding sites has been reported previously, and the results indicate that the alkylation reagents with two reactive sites having the appropriate tether length ($n = 3-5$) induce new single PL peak in SWNTs. Ring forming reactions using the reagents with two binding sites, $I(CF_2)_4I$ and $ICH_2(CF_2)_2CH_2I$, can be proceeded in high efficiency and selectively because the reactive sites are more likely to be encountered intramolecularly. We revised the sentence.

P. 15

‘The effect of the tether length of two binding sites in the reactive alkylation using 1,n-dibromoalkane $(Br(CH_2)_nBr$ has been reported previously, and the results indicate that the alkylation reagents having the appropriate tether length ($n = 3-5$) induce new single PL peak in SWNTs.²⁹ It is known that the small ring-forming reaction is stereoelectronically favorable and has high reaction efficiency. The higher PL selectivity observed in the reaction using $I(CF_2)_4I$ and $ICH_2(CF_2)_4CH_2I$ is due to their suitable tether lengths resulting in higher efficiency of cycloaddition reaction.’

- It was mentioned that “Br(CH2)4Br and I(CF2)4I, with two reactive sites, exhibited higher reactivity than CH3(CH2)3I and CF3(CF2)3I.” However, there was no further discussion on the bromine comparison.

To clarify the effect of halogen, we conducted the reactions using 1-iodobutane and 1-bromobutane. When SWNT- $(CH_2)_3CH_3$ was synthesised with 1-iodobutane and 1-bromobutane, the D/G values were 0.21 and 0.19, respectively, indicating that iodoalkane is relatively higher reactive than corresponding bromoalkane (Table S1). Therefore, we added the sentence in Page 7. The sentence was revised as follow.

P. 7

‘Regarding the effect of halogen atoms at the reactive sites, the D/G values of SWNT- $(CH_2)_3CH_3$ synthesised using 1-iodobutane and 1-bromobutane were 0.21 and 0.19, respectively, indicating that iodoalkane is more reactive than the corresponding bromoalkane.’

- CD was performed on the separated fractions and seem to match results from a previous manuscript by the authors. However, there is no discussion on the relevance of this measurement with respect to this manuscript, and it is therefore unclear how CD adds to the interpretation of the results.

Since the optical resolution of the main component, (6,5) SWNT adducts, was achieved, therefore, the CD spectra was measured and shown in Figure 2. The CD spectra is an important result in understanding the separation behavior of SWNT adducts, therefore, the spectra were shown together with the absorption spectra, as an assignment of the fractions. The sentence was added as follow.

P. 10

The circular dichroism (CD) spectra of these fractions exhibited mirror-image CD spectra similar to those of (11,-5) and (6,5) SWNTs, **indicating** the achievement of optical resolution for the functionalised (6,5) SWNTs (Figure 2c).

- Is there a reason SWNT-CH₂(CF₂)₂CF₃ was not separated using gel chromatography, as the derivatives used for separation were chosen to look at chiralities other than (6,5) and this derivative has the largest amount of (6,4).

In addition to the selective control of PL wavelength, the lowering the emission energy is main topics in this study. In this point of view, separation of SWNT-CH₂(CF₂)₂CF₃ was not rigorously conducted. The observation of (6,4) chiral SWNTs in SWNTs-CH₂(CF₂)₂CF₃ may be due to the moderate functionalization degree of (6,4) SWNTs-CH₂(CF₂)₂CF₃.

4. Clarifications/inconsistencies

- Page 6: there seems to be some inconsistencies with the derivatives presented in the manuscript. Specifically, SWNT-(CH₂)₄ is discussed throughout the results but is not represented in Table S1 nor included in the list of functionalized SWNTs that are being studied.

The results of SWNTs-(CH₂)₄ were added in Table S1, Figure S1, and Figure S2.

Revised Table S1

	SWNT-(CH ₂) ₃ CH ₃	SWNT-(CH ₂) ₃ CF ₃	SWNT-(CH ₂) ₂ C(F ₂)CF ₃	SWNT-CH ₂ (CF ₂) ₂ CF ₃	SWNT-(CF ₂) ₃ CF ₃
E ₁₁ ⁻ PL (nm)	1100	1104	1118	1119	1150
E ₁₁ ⁺ PL (nm)	1231	1240	1243	-	-
D/G _{561nm}	0.19 (Br) 0.21 (I)	0.20	0.18	0.09	0.13
	SWNT>(CH ₂) ₄ ¹	SWNT>CH ₂ (CF ₂) ₂ CH ₂	SWNT>(CF ₂) ₄		
E ₁₁ ⁺ PL (nm)	-	-	-		
E ₁₁ ⁺ PL (nm)	1228	1246	1319		
D/G _{561nm}	0.44	0.06	0.17		

Revised Figure S1

Revised Figure S2

- Page 6: it should be clarified that “the spectral changes after functionalization tended to decrease with an increasing number of fluorine atom substitutions” was concluded from the D/G ratio.

As pointed out by reviewer, the tendency was considered from D/G values. The sentence was revised as follow.

Revised P.6

‘Interestingly, the D/G values after functionalisation increased in the order: [SWNT>CH₂(CF₂)₂CH₂] < [SWNT-CH₂(CF₂)₂CF₃] < [SWNT-(CF₂)₃CF₃] < [SWNT>(CF₂)₄] < [SWNT-(CH₂)₂CF₂CF₃] < [SWNT-(CH₂)₃CF₃] < [SWNT-(CH₂)₃CH₃] < [SWNT>(CH₂)₄]^{22,29}. With the exception of SWNT-CH₂(CF₂)₂CF₃ and SWNT>CF₂(CF₂)₂CH₂, the D/G values after functionalisation tended to decrease with an increasing number of fluorine atoms (Supplementary Table 1).’

- Page 8: It is stated that “... gel chromatography was conducted on SWNT-(CF₂)₄, SWNT-CH₂(CF₂)₂CH₂, and SWNT-(CF₂)₃CF₃”. In Figure 4 and Table 1, which shows the results of the separated SWNTs, SWNT-(CH₂)₄ and SWNT-CH₃(CH₂)₃-SWNT-H (derivatives not listed above) are both shown, but SWNT-CH₂(CF₂)₂CH₂ is not. The derivatives shown in the figures/tables should be consistent with what is described in the manuscript.

Thank you for your precise remarks. The data and plots for SWNT>CH₂(CF₂)₂CH₂ were added to the Figure 4 and Table 1.

Revised Figure 4

Figure 4. Emission energy difference (ΔPL) between the functionalised and non-functionalised SWNTs as a function of the SWNT diameter. SWNT>(CF₂)₄ (blue), SWNT>CH₂(CF₂)₂CH₂ (pink), SWNT>(CH₂)₄ (red)²⁹, SWNT-(CF₂)₃CF₃ (green), and CH₃(CH₂)₃-SWNT-H (black)²⁰.

Revised Table 1**Table 1. PL wavelength of the functionalised SWNTs after separation.**

SWNT	(6,4)	(7,3)	(9,1)	(6,5)	(8,3)	(5,4)	(7,5)
SWNT-(CF ₂) ₄	1064 1222	1373 1389	1207	1320	1269	-	1345
SWNT-CH ₂ (CF ₂) ₂ CH ₂	1003 1139	1288	-	1237	-	-	-
SWNT-(CH ₂) ₄ ²⁹	1115	1265	-	1229	1170	-	1248
SWNT-(CF ₂) ₃ CF ₃	1068	1185	1153	1152	1164	1027	1178
CH ₃ (CH ₂) ₃ -SWNTs-H ²⁰	1015	1128 1213	-	1104 1231	1076	-	1136
SWNTs-(CF ₂) ₅ CF ₃ ²¹	1082	1190	1128	1155	1169	1027	1206

- Page 8: It is stated that "... CF₃(CF₂)₂CH₂I, CF₃(CF₂)₃I, and I(CF₂)₄I produced a single E₁₁* or E₁₁** PL peak, whereas the other reagents produced two PL peaks." However, on page 7 it is stated that "... one prominent PL peak attributed to the functionalised (6,5) SWNTs was observed from SWNT-(CF₂)₃CF₃ (1152 nm), SWNT-CH₂(CF₂)₂CF₃ (1119 nm), SWNT-(CH₂)₄ (1230 nm), SWNT-CH₂(CF₂)₂CH₂ (1246 nm), and SWNT-(CF₂)₄ (1318 nm)." Should SWNT-(CH₂)₄ and SWNT-CH₂(CF₂)₂CH₂ be included on page 8? If not, an explanation as to why they are an exception should be provided.

Thank you for your precise remarks. The SWNTs-(CH₂)₄ and SWNTs-CH₂(CF₂)₂CH₂ exhibited new single PL peak, respectively. The sentence was revised as follows.

Revised P.8

'It is noteworthy that, owing to their high selectivity, CF₃(CF₂)₂CH₂I, CF₃(CF₂)₃I, Br(CH₂)₂Br,²⁹ ICH₂(CF₂)₂CH₂I, and I(CF₂)₄I produced a single E₁₁* or E₁₁** PL peak, whereas the other reagents produced two PL peaks.'

- Page 10: It appears (7,3) SWNT-(CF₂)₄ also exhibited two peaks in the PLE map addition to (6,4). If it should not be included, an explanation as to why should be provided.

As reviewer pointed out, (7,3) SWNT-(CF₂)₄ also exhibited two peaks in the PLE map. The sentence

was revised as follows.

Revised P.10

‘As an exception, (6,4) SWNT > (CF₂)₄ (0.019) and (7,3) SWNT > (CF₂)₄ (0.023) exhibited two PL peaks ((6,4) SWNT: 1064 and 1222 nm and (7,3) SWNT: 1373 and 1389 nm).’

- Page 10: The term “high PL wavelength selectivity” is somewhat confusing. Do the authors mean the derivative favors longer PL wavelengths, or there is high selectivity in the functionalization?

In this context, we focused on the high selectivity in the functionalisation. We revised the manuscript for the clarification.

Revised P.10

‘Unlike functionalisation with bromobutane, which results in two new PL peaks, it is noteworthy that functionalisation using CF₃(CF₂)₃I exhibited high selectivity to emerge a single new PL peak.’

- Page 19: what is meant by “normalized to the measurement time”?

Thank you for your precise remarks. The measurement time was revised to the integration time. The sentence was revised as follows.

Revised P.19

‘The PL intensity was normalized by the integration time.’

Reviewer #3 (Remarks to the Author):

I have reviewed “Selective Emergence of Photoluminescence at Telecommunication Wavelengths from Cyclic Perfluoroalkylated Carbon Nanotubes” by Maeda, et. Al. I believe the results presented are interesting and novel. However, they are iterative on previous efforts to produce SWCNT functionalization that emits in the NIR range. The authors successfully produce a functionalization scheme that generates emission energies of wavelengths longer than 1300 nm. The methods used, however, are a slight variation on previous functionalization schemes that seem to introduce additional electron inducing effects as opposed to generating different configurations or defect isomers. As such, this article is of interest to specialists in the field and has, in my opinion, little general relevance. I think such a report would be well catered for

a publication that focuses on nanotechnology or synthetic methods and would not recommend publication in Communications Chemistry.

Thank you for evaluating our study as interesting and novel. Control of SWNT PL wavelength to the red-shifted region is currently important issue to be used for telecommunications, and the selective emergence of PL over 1300 nm from (6,5) SWNTs is one of the milestones in this fields. In this study, we rationally construct a reaction system based on the factors that control the PL characteristics of SWNTs, resulting in the emergence of the PL at the longest wavelength. We believed the results obtained from current experimental and theoretical studies provide a deep understanding of the relationship between structure and optical properties of SWNTs. We think the obtained results are also useful for understanding the structures and properties of π -conjugated compounds including fullerenes and graphene and the obtained systems are useful for the near-infrared luminescent materials.

REVIEWERS' COMMENTS:

Reviewer #1 (Remarks to the Author):

Thank you for providing corrections and convincing explanations. The manuscript may be accepted for publication.

Reviewer #2 (Remarks to the Author):

Thank you for the the response to my comments and the revisions to the manuscript that help clarify some points. I feel my comments were adequately addressed and recommend publishing the revised manuscript.